# Early resumption of postpartum sexual intercourse and its associated risk factors among married postpartum women who visited public hospitals of Jimma zone, Southwest Ethiopia: A cross-sectional study

**Tariku Bekela Gadisa**[1]*, **Mengistu Welday G/Michael**[2☯], **Mihretab Mehari Reda**[2☯], **Beyene Dorsisa Aboma**[3☯]

1 School of Midwifery, Institute of Health, Faculty of Health Sciences, Jimma University, Jimma, Ethiopia, 2 Department of Midwifery, College of Health Sciences, Mekelle University, Mekelle, Ethiopia, 3 School of Nursing, Institute of Health, Faculty of Health Sciences, Jimma University, Jimma, Ethiopia

☯ These authors contributed equally to this work.
* gadaa2007@gmail.com

**Data Availability Statement:** All relevant data are within the paper and its Supporting Information files.

## Abstract

### Introduction

Postpartum sexual resumption without the use of contraception is a risk for unintended and closely spaced pregnancies. Although counseling related to the resumption of postpartum sexual intercourse is a key component of postpartum sexual health, it is not widely addressed during the postnatal period. Thus, this study aimed to assess the early resumption of postpartum sexual intercourse and its associated risk factors among married postpartum women who visited public hospitals of Jimma zone, Southwest Ethiopia, for child immunization services.

### Methods

The facility-based cross-sectional study design was undertaken, and a systematic random sampling technique was carried out to select 330 participants. Data were collected using a pretested interviewer-administered questionnaire from August to September 2019. Obtained data were analyzed using descriptive statistics. A bivariate analysis was used to determine the significance of the association. Variables that showed association in the bivariate analysis at p-value <0.2 were fitted into a multivariable logistic regression model to control for confounders, and the significance of association was determined at p-value <0.05 with a 95% confidence interval (CI).

### Results

Approximately 53.9% of the respondents practiced early resumption of postpartum sexual intercourse. Factors such as low income (AOR = 0.19 (95% CI = 0.10-.37)), monogamous marriage 3.78(1.32–10.79), practicing sexual intercourse during pregnancy (AOR = 4.55

**Funding:** No funding was received for this work.

**Competing interests:** The authors have declared no competing interest.

(95% CI = 1.29–15.97)), a cesarean delivery (AOR = 0.06 95%CI = (0.03–0.15)) and use of contraceptives (AOR = 3.7(95%CI = 1.92–7.14)) were significantly associated with early resumption of postpartum sexual intercourse.

## Conclusion and recommendation

The findings of this study suggested that, most postpartum mothers resumed sexual intercourse during the early postpartum period and its associated risk factors include low income, monogamous marriage, practicing sexual intercourse during pregnancy, cesarean delivery, and use of contraceptives. Discussion with couples about postpartum sexual health during the antenatal and postnatal period is crucial to prevent unwanted pregnancies and adverse health outcomes.

## Introduction

Postpartum sexual health attributes include the resumption of sexual intercourse, sexual arousal, desire, orgasm, and sexual satisfaction [1]. Resumption of postpartum sexual resumption is defined as having the first penetrative vaginal sexual intercourse after childbirth [2].

Recent evidence showed that the period of postpartum sexual abstinence is decreasing globally. For instance, the result of a recent study conducted in Australia revealed that sexual abstinence for most women ends at 7 weeks of postpartum, showing a shift from the taboo against sexual intercourse after childbirth [3].

A study conducted in Nigeria among postpartum women found that 67.9% resumed sexual intercourse by 8 weeks following childbirth [4], similar to study conducted in Ethiopia which found that 73.4% of women resumed sexual intercourse after childbirth by 6 weeks [5]. Reportedly, the early resumption of postpartum sexual intercourse exposes many women to sexual and reproductive health problems such as sexual discomfort, due to incomplete healing of episiotomy or any lacerations [6, 7].

Consequently, postpartum mothers who resume sexual intercourse too soon after childbirth are at substantially greater risk for infections due to vaginal lesions and abrasions following the labor and delivery process than those who do not practice early resumption of sexual intercourse [6]. In addition, other evidence showed that a majority of mothers who have resumed postpartum sexual intercourse during the first three months typical experienced sexual morbidity such as dyspareunia, lack of vaginal lubrication, difficulty in achieving orgasm, vaginal loosening, lack of sexual desire, abnormal vaginal discharge, and genital tear [4, 6–8].

Furthermore, early resumption of postpartum sexual intercourse might cause unintended pregnancies that may result in numerous poor maternal and child health outcomes if not supplemented with effective contraceptive methods [9, 10].

Unpredictably, a research study conducted in Southeast Nigeria revealed that early resumption of sexual intercourse after childbirth has endangered child health by increasing the incidence of a childhood disease such as fever, diarrhea, measles, and tetanus that culminates in under-five mortality [11].

Factors like spontaneous vaginal delivery [3, 4, 8], a low parity [3, 4], low alive child [3, 4, 12], using contraceptive methods [4, 12], resumption of menses [4, 8], a monogamous marriage [4], and the young age of mothers [3, 4] were some of the factors affecting the early resumption of sexual intercourse among women in the postpartum period.

Though the World Health Organization (WHO) recommends that all women be evaluated regarding the resumption of sexual intercourse as a part of general assessment 2-6weeks following delivery, little attention has been given by researchers, policymakers, and health care providers [13].

Besides, in most developing countries, many postpartum women do not get information or counseling about postpartum sexual health during the antenatal and postnatal period when to resume sexual intercourse safely after delivery [4, 5, 8, 9]. Similarly in Ethiopia, most studies conducted on women's health during the postpartum period focused primarily on family planning utilization [5, 10].

In conclusion, postpartum sexual health is one of the globally emerging agendas since sexual health can be significantly altered during pregnancy, birth, and postpartum [1, 2]. Postpartum counseling for women concerning the early resumption of sexual intercourse, and its associated risk factors remains poorly documented. Hence, this study was aimed to assess the early resumption of postpartum sexual intercourse and its associated risk factors among married women who visited public hospitals of Jimma zone for child immunization services.

## Methods

### Study area, study period, and study design

The facility-based cross-sectional study design was conducted in public hospitals of Jimma Zone, Southwest Ethiopia from August to September 2019.

### Population

The source population of this study was all postpartum women who resumed sexual intercourse and visited public hospitals of the Jimma zone for child immunization services at 14 weeks after childbirth. The study population was postpartum women who resumed sexual intercourse and visited the selected public hospitals of Jimma zone for immunization services at 14th weeks after childbirth.

### Inclusion and exclusion criteria

Married mothers who have resumed sexual intercourse and visited these hospitals for child immunization services at 14 weeks after childbirth were included in the study. Mothers who were critically ill during the study period were excluded from the study.

### Sample size determination

The sample size for this study was determined by using a single population proportion formula with the following assumptions: p = 0.734 from a study conducted in Addis Ababa [5] and confidence interval = 95%, critical value z = 1.96 and degree of precision = 0.05 and considering a 10% non-response rate, the final sample size was 330.

### Sampling procedures

Four out of seven public hospitals, namely; Shenen Gibe, Limmu, Seka, and Omo Nada hospitals were selected randomly by the lottery method. The final sample size was proportionally allocated for the four hospitals based on the previous six-month performance report of postpartum mothers visited in each hospital.

## Sampling technique

The sampling interval of women was determined by dividing the total number of postpartum mothers visited for the previous six months from each hospital by the final sample size that was 4. The first study participant was selected by the lottery method and the subsequent study participants were selected systematically at every fourth interval until the allocated sample size was obtained from each hospital.

## Study variables

**Outcome variable.** Early resumption of postpartum sexual intercourse (resumption of sexual intercourse before six weeks after childbirth).

**Independent variables.** The independent variables of this study include socio-demographic, reproductive and obstetric, sexual health-related variables.

## Operational definitions and definition of terms

**The timing of sexual intercourse resumption.** Was categorized into resumption before 6 weeks (early/unrecommended period) = coded by 1 and after 6 weeks (recommended time) = coded by 0.

**Unrecommended period for sexual resumption.** Vaginal sexual intercourse which occurred within 6 weeks postpartum [1, 7–9, 16, 17].

**Recommended period for sexual resumption.** Vaginal sexual intercourse initiated after 6 weeks postpartum [1, 7, 9, 16, 17].

## Data collection tools and procedures

The questionnaire was prepared in the English language, translated to the Afaan Oromo language (Local language), and then back into the English language to ensure consistency. The questionnaire had both open and close-ended questions and categorized into four major parts:-Socio-demographic factors, obstetric and reproductive factors, postnatal, and sexual health-related factors. The reliability of the tool was checked by cronbach's alpha which was 0.781. This tool also checked for validity by three professional experts and the experts' feedbacks was incorporated into the final questionnaire which was used for the data collection.

Data were collected by structured questionnaires using a face-to-face interview. A two days training was given for data collectors and supervisors of the study and finally, data were collected by four bachelor degrees of science (B.Sc.) holder female midwives and two supervisors working in other health facilities and who have proved experience in data collection.

## Data quality management

During the supervision, the quality, and completeness of gathered information by the data collectors was checked daily by the supervisors, and timely corrections were made and it could help a lot in improving the quality, consistency, and completeness of data for subsequent interviews. A pretest was undertaken on 10% of the calculated sample size before the actual data collection on women visiting immunization services at 14th weeks postpartum in Bedele hospital. Following the result of a pretest inconsistency, skipping problem, and wording ambiguity was seen and modified accordingly.

## Data processing and analysis

Data were entered into the computer using data manager version 4.6 and exported to SPSS version 23 for analysis. And sexual intercourse resumption was coded by 1 for mothers who

practiced sexual intercourse resumption before six weeks and a code by 0 for those who practiced sexual intercourse resumption after six weeks (recommended time).

Bivariate logistic regression analysis was used to show the significance of the association. The variable that showed the significant association in a bivariate analysis at p-value ≤ 0.2 [5] was entered into multivariable logistic regressions to control for confounding and the significance of association was determined at 95% confidence interval and p-value <0.05. The Hosmer-Lemeshow statistic p-value had a chi-square value of 10.3 and a significance level of 0.174 thus, the model is fit. Multi-collinearity was checked interaction among independent variables by a variance inflation factor (VIF) which was less than ten.

### Ethical approval

Ethical clearance was obtained from the Institutional Review Board (IRB) of Mekelle University, College of Health Sciences, and brought to the Jimma zone Health Bureau and from there to respective selected hospitals. The data collectors gave detailed explanations on the purpose of the study, clarified that participation was voluntary. The informed written consent was obtained from each participant before the interview. In addition; informed written consent was taken from the parents or the legal guardian for the participants who were less than 18 years of age. The privacy of the participants was secured by interviewing them in a private classroom. The information provided by each respondent was kept confidential and de-identified.

## Results

### Socio-demographic characteristics

Of the total 330 sampled postpartum women, 319 participants have completed the interview and making a response rate of 96.7%. The mean age of participants was 27.88 years (SD ± 6.157) and About 277(86.8%) of the participants were having a monogamous relationship and 42(13.2%) were polygamous relationships. About 199(62.4%) participants were from urban residence and 120(37.6% were from rural residence and 198 (62%) participants have family monthly income of below or equal to the mean 1532 ETB and 121(37.9%) participants have above the mean (Table 1).

### Obstetric and reproductive related characteristics

According to this study, the mean parity of mothers was 2.76(SD±2.32), and the mean living children were 2.79(SD± 2.37) and more than half of participants had 104(55.3%) interpregnancy intervals in less than two years. About 53 (28.2%) of participants reported a history of abortion and of this nearly half, 46.8 (86.8%) reported history of one abortion (Table 1).

### Postnatal related characteristics

The study found that 287 (90%) of participants had no PNC visit whereas 32(10%) had postnatal care visit and concerning infant feeding practice, more than four-fifth 274(85.9%) of participants were practicing exclusive breastfeeding and 45 (14.1%) were practiced formula feeding (Table 2).

### Sexual health-related characteristics

**Sexual intercourse during the index pregnancy.** Among the participants, 285 (89.3%) had practiced sexual intercourse during the last pregnancy and of this nearly half 137(48.1%) of them ends sexual intercourse between 28–37 weeks of gestation (Table 2).

**Table 1. Distribution of socio-demographic, reproductive and obstetric related characteristics among married postpartum women at the selected four Jimma zone Public Hospitals, August 15 to September 15, 2019.** N = 319.

| Variables | Category | Frequency | Percent |
|---|---|---|---|
| Age | 15–19 | 21 | 6.6 |
| | 20–24 | 78 | 24.5 |
| | 25–29 | 105 | 32.9 |
| | 30–34 | 58 | 18.1 |
| | 35–49 | 57 | 17.9 |
| Ethnicity | Oromo | 270 | 84.6 |
| | Dawuro | 17 | 5.3 |
| | Gurage | 12 | 3.8 |
| | Kefa | 13 | 4.1 |
| | Others* | 7 | 2.2 |
| Religion | Muslim | 268 | 84 |
| | Orthodox Christian | 33 | 10.4 |
| | Protestant Christian | 11 | 3.4 |
| | Catholic | 7 | 2.2 |
| Educational status | No formal education | 83 | 26.0 |
| | Primary(1–8) | 49 | 15.4 |
| | Secondary(9–12) | 95 | 29.8 |
| | Diploma/Degree | 92 | 28.8 |
| Occupation | Housewife | 185 | 58.0 |
| | Private /self employed | 50 | 15.7 |
| | Government employed | 55 | 17.2 |
| | Merchant | 15 | 4.7 |
| | Others** | 14 | 4.4 |
| Duration of living together with husband | <2 | 102 | 32.0 |
| | 2–4 | 60 | 18.8 |
| | >/ = 5 | 157 | 49.2 |
| Parity | Primipara | 155 | 48.6 |
| | Multipara | 88 | 27.6 |
| | Grand multipara | 76 | 23.8 |
| Number of alive children | 1–2 | 207 | 64.9 |
| | 3–4 | 36 | 11.3 |
| | >/ = 5 | 76 | 23.8 |
| Interpregnancy interval | <24 | 104 | 55.3 |
| | 24–47 | 49 | 26.1 |
| | ≥48 | 35 | 18.6 |
| Index pregnancy status | Planned | 249 | 78.1 |
| | Un planned | 70 | 21.9 |
| ANC follow up for the last pregnancy | Yes | 285 | 89.3 |
| | No | 34 | 10.7 |
| ANC number | 1–3 | 82 | 28.8 |
| | >/ = 4 | 203 | 71.2 |
| Obstetric complication during last pregnancy | Yes | 51 | 16 |
| | No | 268 | 84 |
| Type of complication | APH | 18 | 35.3 |
| | Hypertensive disorders | 16 | 31.4 |
| | PROM | 12 | 23.5 |
| | Other | 5 | 9.8 |

*(Continued)*

**Table 1.** (Continued)

| Variables | Category | Frequency | Percent |
|---|---|---|---|
| Place of birth | Health center | 77 | 24.1 |
| | Governmental hospital | 214 | 67.1 |
| | Private clinic/Hospital | 15 | 4.7 |
| | Home | 13 | 4.1 |
| Labor duration in hrs. | <6 | 90 | 28.2 |
| | 6–12 | 160 | 50.2 |
| | >12 | 69 | 21.6 |
| Mode of birth | Vaginally delivery | 257 | 81% |
| | Cesarean delivery | 62 | 19% |

NB: * = Silte, Yem, Amhara and

** = Student, daily laborer and

*** = Multiple pregnancy and Anemia

**Early resumption of postpartum sexual intercourse.** This study revealed that the earliest period of sexual resumption was week two and the latest was 13 weeks. The mean time of sexual resumption was 6.7 weeks (SD≤ 2.25). Overall total participants 319,172(53.9%) were resumed sexual intercourse during the unrecommended time (</ = 6 weeks postpartum) and of this about 45 (14.1%) of mothers were resumed sexual intercourse within 4 weeks of delivery (Table 2). About three fourth 244 (76.5%) of participants were reported that the reason for resumption was requested by their husbands (Table 2).

## Factors associated with early resumption of postpartum sexual intercourse

Variables having a p-value less than or equal to 0.2 on binary logistic regression was a candidate for multivariable logistic analysis, these variables were household income, wife number, nature of index pregnancy(planned/unplanned), the practice of sexual intercourse during the index pregnancy, mode of birth, infant feeding practice, menstrual resumption pattern and using contraceptive methods.

In multivariable logistic regression husband-wife number, household monthly income, the practice of sexual intercourse during the index pregnancy, mode of birth, and using contraceptive methods were found to be significantly associated with resumption of sexual intercourse during the unrecommended period at a p-value less than 0.05 with 95% confidence interval.

This study revealed that, those study participants whose average household monthly income </ = 1532 ETB were reduced odds by 80.9% to resume sexual intercourse before six weeks than mothers whose monthly income greater than 1532ETB (AOR = 0.19, 95% CI (0.10-.37).

Mothers who practiced sexual intercourse during pregnancy were increased odds by 4.547 to resume sexual intercourse during an unrecommended period when compared to mothers who did not practice sexual intercourse during pregnancy (AOR = 4.55, 95% CI = 1.29–15.97).

Mothers who undergo cesarean delivery were reduced odds by 93.5% to resume sexual intercourse during an unrecommended period than mothers who delivered by vaginal delivery (AOR = 0.06, 95% CI = 0.03–0.15).

Similarly, women from a husband with one wife was around four times more likely to resume sexual intercourse during the unrecommended period than mothers from a husband who has more than or equal two wives (AOR = 3.78, 95% CI = 1.32–10.79) and using

**Table 2. Distribution of postnatal, sexual health related characteristics among married postpartum women who were attending immunization centers at the selected four Jimma zone Public Hospitals, Jimma, Southwestern Ethiopia, August 15 to September 15, 2019.** N = 319.

| Variable | Category | Frequency | Percent |
|---|---|---|---|
| Postpartum related complications | Yes | 23 | 7.2 |
| | No | 296 | 92.8 |
| Type of postpartum complication reported | PPH | 12 | 52.2 |
| | Hypertensive disorder | 6 | 26.1 |
| | Puerperal sepsis | 5 | 21.7 |
| Knowledge of at least one contraceptive method | Yes | 289 | 90.6 |
| | No | 30 | 9.4 |
| Which contraceptive methods you did you know? | Pills | 153 | 20.4 |
| | Injectable | 279 | 37.2 |
| | Implants | 219 | 29.2 |
| | IUCD | 79 | 10.5 |
| | Others***** | 20 | 2.7 |
| Are you using contraceptive method currently? | Yes | 184 | 57.7 |
| | No | 135 | 42.3 |
| Which method are you currently using? | Pills | 10 | 5.4 |
| | Injectable | 132 | 71.7 |
| | Implants | 34 | 18.5 |
| | IUCD | 8 | 4.3 |
| Practice of sexual intercourse during index pregnancy | Yes | 285 | 89.3 |
| | No | 34 | 10.7 |
| Weeks last sexual intercourse ends during index pregnancy | <28 | 74 | 26.0 |
| | 28–37 | 137 | 48.1 |
| | >/ = 37 | 74 | 26.0 |
| Early resumption of sexual intercourse | Yes | 172 | 53.9 |
| | No | 147 | 46.1 |
| Reason given for resumption of intercourse | Husband demand sex | 244 | 76.5 |
| | Jointly decision for sexual intercourse resumption | 54 | 16.9 |
| | Others ****** | 21 | 6.6 |

Other ***** = Female sterlization, condom,

****** = cultural demand, fear of marital disharmony, felt convenient

contraceptive the method has around four times 3.7 more likely to resume sexual intercourse during the unrecommended period than mothers who did not use contraception (AOR = 3.7, 95% CI = 1.92–7.14) (Table 3).

## Discussion

Evidence from this study revealed that half of participants 172(53.9%) had resumed sexual intercourse before six weeks (during the unrecommended time) which was almost similar to the study reported from Malaysia, 51.6% [9]. However, this result was higher than earlier reports of most countries like the USA (43%), India (28.3%), Nigeria (27.6%), and Uganda (21, 9%) [12, 14, 15, 16] and Ogbomoso, Nigeria, 40% [17].

And this result was again lower compared to figures reported from Poland (72.8%), Brazil (70%) [18, 19], and Addis Ababa 73.4% [5]. The discrepancy in the time of resumption of postpartum sexual intercourse might be due to diverse socio-economic that exists in low, middle, and high-income countries and also diverse sexual attitudes of women in different parts of the world.

**Table 3. Factors associated with early resumption of postpartum sexual intercourse from August 15 to September 15, 2019 n = 319.**

| Variables | Category | Early postpartum sexual resumption | | COR | AOR |
|---|---|---|---|---|---|
| | | Yes | No | | |
| Pregnancy index | Planned | 141(56.6%) | 108(43.4%) | 1.64(.96–2.8)* | 0.95(0.44–2.03) |
| | Unplanned | 31(44.3%) | 39(55.7%) | 1 | 1 |
| Form of marriage | 1 | 166(59.9%) | 111(40.1%) | 8.97(3.66–22.04)* | 3.78(1.32–10.79)** |
| | >/ = 2 | 6 (14.3%) | 36 (85.7%) | 1 | 1 |
| Infant feeding practice | Exclusive breastfeeding | 151(55.1%) | 123(44.9%) | 1.4(.746–2.64)* | 0.81(0.33–1.99) |
| | Formula feeding | 21(46.7%) | 24(53.3%) | 1 | 1 |
| Family monthly income | </ = 1542 ETB | 77(38.9%) | 121 (61.1%) | 0.17(0.1–0.29)* | 0.19 (0.1–0.37)** |
| | >1542 ETB | 95(78.5%) | 26(21.5%) | 1 | 1 |
| The practice of sexual intercourse in pregnancy | Yes | 167(58.6%) | 118(41.4%) | 8.21(3.09–21.83)* | 4.55 (1.29–15.97)** |
| | No | 5(14.7%) | 29(85.3%) | 1 | 1 |
| Mode of birth | Vaginally | 161(62.6%) | 96(37.4%) | 1 | 1 |
| | Cesarean | 11(17.7%) | 51(82.3%) | 0.13(0.06–0.26)* | 0.06(0.03-.15)** |
| Using contraceptive | Yes | 128(69.6%) | 56(30.4%) | 4.73(2.93–7.62)* | 3.7 (1.92–7.14)** |
| | No | 44(32.6%) | 91(67.4%) | 1 | 1 |
| Resumption of menses | Yes | 100(58.8%) | 70(41.2%) | 1.528(0.981–2.380)* | 0.79 (0.43–1.46) |
| | No | 72(48.3%) | 77(51.7%) | 1 | 1 |

NB: 1 = Reference

* = p-value ≤ 0.2 and

** = p-value < 0.05

It is interesting to note that, the slight decrease from the study done in Addis Ababa in the same locality compare to this study [5].

Maybe because, as the study from Addis Ababa [5] was done in the capital city of the country, where peoples from a lot of countries living and which may share cultural transformations, modernizations, and social changes among peoples which can affect an individual's perceptions and behavior mostly on sexuality issue. Whereas, these study settings in which most of the women adhere to the preexisting culture.

Similarly, this study revealed that the earliest time for sexual intercourse resumption was on week two, this finding is supported by research reported from Kenya [20], the earliest sexual resumption was week two, the possible reason might be similarity in due sharing some preexisting socio-cultural context where the study was conducted. But earlier than studies reported from Malaysia (7 weeks), Nigeria (8 weeks), and Poland (10 weeks) [9, 17, 18] and these discrepancies might be since postpartum sexuality counseling received from health care providers better than in this study.

From the multivariable regression analysis, the monogamous form of marriage was one of the significant factors of sexual intercourse resumption during the unrecommended period. This study revealed that women with monogamous marriage were nearly four times more likely to resume sexual intercourse during the unrecommended postpartum period than mothers with polygamous marriage(AOR = 3.779, 95% CI = 1.323–10.794). This study is supported by a study done in Nigeria [8].

This might be justified that a husband who had a polygamous form of marriage can create an environment to have sexual intercourse with one of the wives when the other one was abstaining.

But recently due cultural transformation, modernization, and the introduction of modern contraceptives, polygamous marriage is decreasing globally which increases the resumption of sexual intercourse earlier, unlike in the past [3, 8].

This study also revealed that those study respondents' average household monthly income $</ = 1532$ ETB were reduced odds by 80.9% to resume sexual intercourse during the unrecommended time when compared to mothers whose monthly income >1532 ETB(AOR = .191, 95% CI (.10-.367). The finding of this study is in line with the study reported from Malaysia, Nigeria [9, 15]. This might be because relatively women who had better income may need more children and those with low income may not want to have more children beyond their family income.

This study also found that mothers who practiced sexual intercourse during pregnancy were 4.547 times raised odds of resuming sexual intercourse during unrecommended time than mothers who did not practice sexual intercourse during the last pregnancy (AOR = 4.547:95% (CI = 1.295–15.972). This finding is consistent with the study reported from China [21], this might be explained that most of the time the husband was the main initiator of sexual activity and so women were forced to resume soon childbirth to satisfy their husband's demand.

Mothers who undergo cesarean delivery was reduced odds by 93.5% to resume sexual intercourse during an unrecommended period than mothers who delivered by vaginal delivery (AOR = .065, 95% CI = .028 - .153). This study finding was consistent with the study reported from Australia, Kano Nigeria, Southwest Nigeria [3, 4, 17] and the possible reason might be the recovery process associated with a cesarean section is longer than vaginal delivery, because women may have more pain and discomfort in her abdomen as the skin and nerves surrounding her surgical scar need a long time for healing.

Finally, this study found that those postpartum women who were using contraceptive methods at the time of the study were 3.704 times more likely to resume sexual intercourse during unrecommended time than mothers who did not use contraceptive methods (AOR = 3.487, 95% CI = 1.920–7.145). This study is consistent with the study figures reported from Malaysia, USA, Uganda, Nigeria, Malawi [9, 12, 16, 17, 22]. This might be because when women using contraceptives, they consider themselves to be no risk for pregnancy only, which motivates them to resume sexual intercourse before six weeks postpartum. But early sexual intercourse resumption may increase the chance of postpartum sexual morbidity [5] and incomplete healing of episiotomy/tear (if any) in which the open wound is susceptible to infection [6].

## Limitation of the study

Mothers who had a most recent stillbirth were not included in this study since this study was done among mothers who came for vaccination of their child. The study mainly focused on the period of sexual intercourse resumption rather than considering the degree of postpartum sexual function. Being only a facility-based study, recall bias and not approaching women at other MCH services was also among the limitation of this study. Finally, a causal relationship could not be assessed due to the cross-sectional design of the study.

## Conclusions

This study revealed that a high number of respondents were resumed postpartum sexual intercourse during the unrecommended time (the early postpartum period) and this finding, in addition to those of other studies, postpartum resumption of sexual intercourse during unrecommended time is significantly associated with factors such as low income, a monogamous

form of marriage, practicing sexual intercourse during pregnancy, cesarean delivery, and contraceptive usage.

## Recommendations

Postpartum sexuality counseling should be strengthening during the antenatal and postpartum periods. Postpartum women should receive adequate sexuality counseling before leaving the health facility and Additional studies using qualitative study designs may be useful to explore further findings concerning the internal feelings of women regarding postpartum sexual intercourse resumption and the reason why women in this setting had practiced postpartum sexual intercourse during the unrecommended period.

## Acronyms / abbreviations

ACOG, the American College of Obstetrics and Gynecology; AIDS, Acquired Immune Deficiency syndrome; APH, Ante Partum Hemorrhage, Central Statistical Agency; EDHS, Ethiopian Demographic and Health Survey; IUCD, Intrauterine contraceptive device; JUMC, Jimma University Medical Center; SDG, Sustainable Development Goal; SPSS, Statistical Package for Social Science; WHO, World Health Organization.

## Supporting information

**S1 Questionnaire. Questionnaire in both English and local version.**
(DOCX)

**S1 File. Consent form.**
(DOCX)

**S1 Data.**
(SAV)

## Acknowledgments

The authors would like to pass their appreciation to Mekelle University for the approval of ethical clearance and other necessary support. We would like to extend our thanks to the Jimma zone health office, the Medical Directors of each hospital, data collectors, and each participant for their unreserved support and cooperation in providing us data and other necessary preliminary information.

## Author Contributions

**Conceptualization:** Tariku Bekela Gadisa, Mengistu Welday G/Michael, Mihretab Mehari Reda.

**Data curation:** Tariku Bekela Gadisa, Mengistu Welday G/Michael, Beyene Dorsisa Aboma.

**Formal analysis:** Tariku Bekela Gadisa.

**Funding acquisition:** Tariku Bekela Gadisa.

**Investigation:** Tariku Bekela Gadisa.

**Methodology:** Tariku Bekela Gadisa, Mengistu Welday G/Michael, Mihretab Mehari Reda, Beyene Dorsisa Aboma.

**Project administration:** Tariku Bekela Gadisa.

**Resources:** Tariku Bekela Gadisa.

**Software:** Tariku Bekela Gadisa, Mihretab Mehari Reda.

**Supervision:** Tariku Bekela Gadisa, Mengistu Welday G/Michael.

**Validation:** Tariku Bekela Gadisa.

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
