## [Decision Letter · Decision Letter 0]

18 Sep 2020

PONE-D-20-25368

Early resumption of post partum sexual intercourse and its associated factors among married post partum women at Jimma Zone public Hospitals,Jimma ,South West Ethiopia :The cross sectional study.

PLOS ONE

Dear Dr. Gadisa,

Thank you for submitting your manuscript to PLOS ONE. After careful consideration, we feel that it has merit but does not fully meet PLOS ONE’s publication criteria as it currently stands. Therefore, we invite you to submit a revised version of the manuscript that addresses the points raised during the review process.

We look forward to receiving your revised manuscript.

Kind regards,

Nülüfer Erbil, Ph.D, Prof.

Academic Editor

PLOS ONE

Journal Requirements:

2. Thank you for including your ethics statement:  "Ethical clearance and support letter was obtained from the Institutional Review Board (IRB) of Mekelle University, College of Health Science which was brought to Jimma Zone Health Bureau and from there to respective selected Hospitals, finally permission was ensured from each participants."

Please provide additional details regarding participant consent. In the ethics statement in the Methods and online submission information, please ensure that you have specified what type you obtained (for instance, written or verbal, and if verbal, how it was documented and witnessed). If your study included minors, state whether you obtained consent from parents or guardians. If the need for consent was waived by the ethics committee, please include this information.

3. Please state whether you validated the questionnaire prior to testing on study participants. Please provide details regarding the validation group within the methods section.

Reviewers' comments:

Reviewer's Responses to Questions

**Comments to the Author**

1. Is the manuscript technically sound, and do the data support the conclusions?

Reviewer #1: Partly

Reviewer #2: Yes

2. Has the statistical analysis been performed appropriately and rigorously? 

Reviewer #1: Yes

Reviewer #2: Yes

3. Have the authors made all data underlying the findings in their manuscript fully available?

Reviewer #1: No

Reviewer #2: Yes

4. Is the manuscript presented in an intelligible fashion and written in standard English?

Reviewer #1: No

Reviewer #2: No

5. Review Comments to the Author

Reviewer #1: The authors tried to address a gap on postpartum sexual health. That is interesting.

Here are my concerns

1. Generally, it has many editorial problems: grammatical, wording, sentence word agreement.

Title Page:

2. The title is too long catch its concept therefore make short up to 15 words.

3. Both of the following individuals were listed on acknowledgment and authors. Generally, you put them on acknowledgment list means; they do not meet authorship criteria. So, we need strong justification why they included as authors.

i. Dr. Mengistu Welday

ii. Mr.Mihretab Mehari

Abstract:

1. Method: Every four intervals how? It is obvious that the study participants came randomly and how do you know whether the mother is on fourth or not?

2. Result: Your sample: 319, Your respondents: 319,172: so from where you got this?

Body of the manuscript

Introduction

1. Totally it requires rearrangement of paragraphs and sentences. For example paragraph 2, 3 and 4 talks about its consequence, the rest paragraph talks about what early resumption of sexual intercourse mean.

Rearrange like this, definition, start with the global outlook, you can continue with consequence and at the end the reason why you conduct this study.

2. Remove, “Additionally, Early intercourse sexual resumption to past, men were generally married to several wives and could have sexual intercourse with the other wives, whereas the other wife was obliged to stay away from the husband to offer her opportunity to the breast for as long as 2-3 years without intercourse to prevent the pregnancy [9].”…Unrelated to your topic of interest. This about marriage not sexual resumption.

Method

1. Your study populations are mothers who came for vaccination. What about mothers who came for others services a mother of 10 wks and came for outpatient department services, other MCH services?

2. Avoid redundancy example: The married postpartum women who already resumed sexual intercourse and on their 14 weeks

3. “Knowledge of modern contraceptive methods: When a woman spontaneously mentions at least one of the modern contraceptives. She was considered as knowledgeable [14].” What is the importance of this? Are you going to address about knowledge of family planning. Remove this

4. In some places you used 14th weeks for time of the event, and in other places you mentioned six weeks. Make it consistent either use 14thweeks or 6weeks.

5. Sampling technique is totally not clear and invisible for me. You don’t know the mother who is on 1st, 2nd etc.. so how do you reached?

6. 6A. “The questionnaire was prepared in the English language then translated to the Afaan Oromo language (Local language)” and again.

6B. “Data were collected by structured questionnaires using a face-to-face interview which was adopted from related published articles and modified for the current study.”

Confusing, make it clear. Have you prepared? For what purpose? if your data is collected with the adopted instrument correct.

Result

1. “Overall total participants 319,172(53.9%) were resumed …. 319 sample vs 319,172 respondent. Based on this it is difficult to look all result. I have doubt.

2. How do you use mean for parity and living children? Is it continues variable? More clarification

Discussion

1. Your discussion almost seems copy of result. So let you see and amend it. Because no neeed of copy pasting what was available in the result. Start with your result, interpret , compare with other and put what does it implies.

2. Avoid redundancy. Example 1st paragraph: According to this study, about 172(53.9%) Second pargraph : Evidence from this study revealed that half of participants 172(53.9%).

a. What is the implication of this result?

3. Socio-economic vs sexual resumption. How do you relate it?

4. “sexual intercourse resumption was 2 and 13”..how do you give justification based on study period?

a. No way even your comparison groups are not from similar background people. Therefore modify your justification

Limitation of the study

The design you used by itself is also an another limitation of this study

Recommendation: you forwarded as postpartum counseling on sexual health. But 90% of your study participants have no any PNC visit. Did your recommendation and finding go together?

ACKNOWLEDGMENTS

Requires modification. Because those who were acknowledged were put here under acknowledgment

Reviewer #2: Abstract

Regarding the variables that were related to the early onset of sexual intercourse after childbirth, it can be concluded that these variables are considered as a risk factor and to prevent adverse consequences, early onset in people who have a risk factor. Consider and give these people special knowledge and advice.

Introduction

- Paragraphs 2, 3 and 4 can be merged into one paragraph. Modify the whole article in terms of paragraphing. The manuscript was edited by a native speaker.

- These sentences to be cited "And most of the published articles were reported only on the timing of postpartum sexual intercourse resumption without considering its associated factors even though, knowing associated factors of postpartum sexual intercourse resumption is an important element to take an intervention".

- At the end of discussion, it is necessary to imply the studies with similar and different results.

- The difference between the research community conducted in this study and the connection of the Ethiopian country should be noted. So the writers can show the importance of the issue better.

Method

- In the abstract, the type of study is mentioned the Institution-based cross-sectional and in the method section of the manuscript, it is facility-based cross-sectional.

- Some information in population part and Eligibility criteria part are repetitive.

- "Operational definitions and definition of terms" It is usually not available in the journal format. This explanation can be fully explained in the inclusion and exclusion criteria. Assigning codes to variables can be explained in the analysis section.

- In the section on ethical considerations, it is necessary to mention the code of ethics.

Results

- Descriptive data (demographic, fertility and postpartum) can be summarized in one table and the other table devoted to analytical data. It is better to explain in the table and report the adjusted and unadjusted. The number of tables is a lot.

Discussion

- These sentences to be cited "It is interesting to note that, the slight decrease from the study done in Addis Ababa in the same locality compare to this study."

- There is not recommendation for future study. This sentence is not limitation of study " lacking qualitative part of study design was other the limitation of this study because mother’s potential experience on intercourse resumption by culture, norms, and myths related factors were not exhaustively included".

- One of the limitations of the study was .facility base and non-random sample selection, which needs to be mentioned.

6. PLOS authors have the option to publish the peer review history of their article (what does this mean?). If published, this will include your full peer review and any attached files.

Reviewer #1: No

Reviewer #2: No

---

## [Author Response · Author response to Decision Letter 0]

2 Oct 2020

Article Title: Early resumption of sexual intercourse and associated factors among married postpartum women at Jimma zone Public Hospitals, South West Ethiopia: A cross-sectional study.

1) To the academic editor. Thank you, Prof., for your smart feedback. Here are our responses to your points.

• The PLOS ONE style templates: The final manuscript has modified by using PLOS ONE style linkage 

• Ethics statement: Informed written consent was obtained from each participant before the interview. The parents or the legal guardian written consent was taken for the participant age less than 18 years old.

• Validated the questionnaire: the validity of the questionnaire was checked by senior professional experts of Maternal and Child health. The questionnaire was given to three professional experts, and they saw the whole tool and provided their input. Finally, the input provided was incorporated into the final version of the questionnaire that was used for data collection. 

• ORCID iD: This my ORCID ID https://orcid.org/0000-0002-8132-8678

• Supporting files of this study was included in the submission of the manuscript and again revised.

1. Response to Reviewer One 

First, we want to appreciate your smart comments and suggestions, and said these here are our concerns and with some clarification on comments given:

1. Generally comments on grammar ,spelling and subject verb agreement= revised 

2. Title modification=Early resumption of sexual intercourse and associated factors among married postpartum women in Jimma Zone Public Hospitals, Southwest, Ethiopia.

3. Concerning Dr. Mengistu Welday & Mr.Mihretab Mehari both helped me throughout my thesis and manuscript work and mistakenly written under the acknowledgment part. And both of them fit preset criteria of PLOS ONE journal for authorship and are written as authors of this article.

Abstract sections

1. Methods .The study populations were 

Only mothers you have resumed sexual intercourse and came for vaccination of their child at 14th weeks of postpartum. 

First, four mothers waited, then from four mothers one mother was selected by lottery method, then after the first mother was selected, the subsequent every four mothers (since k=4) were included until the desired sample size was obtained from each Hospital.

2. Result: The total sample was 330. From 330 samples, 319 were participated in the study making a response rate of 96.7%: From 319 participants who resumed sexual intercourse, 172(53.9%) resumed sexual intercourse before six weeks or during the early postpartum time and 147(46.1%) resumed after six weeks (during the safe time for resumption).

Body of the manuscript sections

1. The study populations were married postpartum mothers came for vaccination of their child at 14 weeks postpartum (14 weeks postpartum was selected to access all the postpartum women )

2. Study participants also women who resumed sexual intercourse after their recent birth.

3. Mothers who came on 6th, 10th -week vaccination, outpatient, and MCH service were not reached by this study.14th postpartum was selected to access all postpartum women came for vaccination because it is 3rd Ethiopia pediatrics immunization schedule(the vaccination schedule is 1st at 6 weeks postpartum, 2nd at 10th weeks postpartum, and 3rd at 14th weeks postpartum).

4. Redundancy issue appropriate modification was done accordingly

5. Knowledge of modern contraceptive methods: Removed since unrelated to the subject matter.

6. This study was done on mothers visited for vaccination of their child at 14 weeks. The time of sexual resumption was categorized into 6 weeks and below represents early postpartum sexual resumption, and after 6 weeks represent as safe sexual resumption period (the recommended period). 

7. The issue on sampling technique: In the beginning, four mothers who resumed sexual intercourse waited then one mother was selected by the lottery method, and then systematic random sampling at every four intervals was applied until the desired sample size was obtained at each hospital.

8. The questionnaires were prepared by the co-author by reviewing related published articles .Afaan Oromo (local language) version was used for the collection of the data.

Result

1. The total sample size was 330 and of this 319 was responded to the questioner. From 319 participants, 172 participants were resumed sexual intercourse during the early postpartum period (</=6weeks) and 147 participants were resumed during a safe period (> 6weeks).

2. Parity and living children: We asked in terms of a number e,g.How many living children you have? ___in number: In this case, it is a continuous variable for this reason, we used mean and SD before dummy coded into categorical data on SPSS.

Discussion

1. Result and Discussion overlapping issue has amended. 

2. The redundancy issue amended accordingly, and 53.9 = implies that half of the mothers resumed sexual intercourse during the early postpartum period.

3. Socioeconomic vs sexual resumption: From the findings of this study and also other study findings, socioeconomic variables have an association with early sexual intercourse resumption. For instance, mothers who have better income can have an early resumption of sexual intercourse than mothers who have low income.

4. 2 and 13 weeks justification. Both countries share some preexisting socio-cultural context among the continent. 

Limitation

1. Yes, the cross-sectional study design was also among the limitation= modified.

 Recommendations

1. My recommendation is not only for mothers during a postnatal visit rather starting sexual health counseling during ANC visit and strengthening during the early postpartum period (better if it is given before leaving health facilities after birth).

Acknowledgments

1. 1) Amended

 Thank you very much.

 With best regards!

1. For the second reviewer & many thanks for your suggestions, comments, and concerns

Here are a response to your comments and concerns 

1. The knowledge and advice gap on postpartum sexual health has been seen and modified.

So, the women must get adequate knowledge and advice on postpartum sexual health during the antenatal and postnatal period.

1. Paragraph 2, 3, and 4 one paragraph=the introduction section are rearranged and this paragraph also merged into one paragraph.

Methods

1. And most of the published articles were reported only on the timing of postpartum sexual intercourse resumption without considering its associated factors even though, knowing associated factors of postpartum sexual intercourse resumption are an important element to take an intervention". Cited by Reference No [5] and revised in the new paragraph. 

2. The type of this study is facility-based cross-sectional and the consistency issue is corrected.

3. Code of ethics: the study participants were informed about the purpose, risks, and benefits of this study. And voluntary participation and no relation with the service given whether she participated in the study or not. Data collections were conducted after getting written informed consent from each participant. And for the participants aged below 18 years (to fit preset WHO criteria of the age) informed written consent was obtained from the parents or legal guardians.

4. Operational definitions and definitions of terms are omitted and written under the eligibility criteria and analysis section.

Results

1. Table:1 & 2 merged into Table :1 ,Table:3 & 4=merged in to Table=2 and Table :5 to Table:3

2. “It is interesting to note that, the slight decrease from the study done in Addis Ababa in the same locality compare to this study." Cited by Reference No [5]

3. The limitation of the study. “Lacking qualitative part of study design was other the limitation of this study because mother’s potential experience on intercourse resumption by culture, norms, and myths related factors were not exhaustively included”. Omitted and revised into being only a facility-based study was also among the limitation of this study. Finally, a causal relationship could not be assessed due to the cross-sectional design.

Thank you very much.

 With best regards!

---

## [Decision Letter · Decision Letter 1]

4 Jan 2021

PONE-D-20-25368R1

Early resumption of  sexual intercourse and associated factors among married post partum women at Jimma Zone public Hospitals,Jimma ,South West Ethiopia :The cross sectional study.

PLOS ONE

Dear Dr. Gadisa,

Thank you for submitting your manuscript to PLOS ONE. After careful consideration, we feel that it has merit but does not fully meet PLOS ONE’s publication criteria as it currently stands. Therefore, we invite you to submit a revised version of the manuscript that addresses the points raised during the review process.

We look forward to receiving your revised manuscript.

Kind regards,

Nülüfer Erbil, Ph.D, Prof.

Academic Editor

PLOS ONE

Reviewers' comments:

Reviewer's Responses to Questions

**Comments to the Author**

1. If the authors have adequately addressed your comments raised in a previous round of review and you feel that this manuscript is now acceptable for publication, you may indicate that here to bypass the “Comments to the Author” section, enter your conflict of interest statement in the “Confidential to Editor” section, and submit your "Accept" recommendation.

Reviewer #1: All comments have been addressed

Reviewer #2: All comments have been addressed

Reviewer #3: (No Response)

2. Is the manuscript technically sound, and do the data support the conclusions?

Reviewer #1: Yes

Reviewer #2: Yes

Reviewer #3: Yes

3. Has the statistical analysis been performed appropriately and rigorously? 

Reviewer #1: Yes

Reviewer #2: Yes

Reviewer #3: Yes

4. Have the authors made all data underlying the findings in their manuscript fully available?

Reviewer #1: Yes

Reviewer #2: Yes

Reviewer #3: Yes

5. Is the manuscript presented in an intelligible fashion and written in standard English?

Reviewer #1: No

Reviewer #2: Yes

Reviewer #3: No

6. Review Comments to the Author

Reviewer #1: Generally, the Authors made a substantial improvement for the previous comment. What was left is editorial problem and I do have only few concern for clarification and what the authors have to enrich for the article.

1. Answer for 6 states that, “This study was done on mothers visited for vaccination of their child at 14 weeks. The time of sexual resumption was categorized into 6 weeks and below represents early postpartum sexual resumption, and after 6 weeks represent as safe sexual resumption period (the recommended period). Therefore if early sexual resumption is less than six (6) weeks, how do you say early sexual resumption for mothers who came at 14th week?

2. Your response also indicated that mothers who came at 6weeks, 10th weeks and mothers visiting other MCH services were not included. So you can state this under limitation.

Methods

3. There are many sub sections but contents under some of the subsection are not as much sound. It needs some modifications and merging

Result

4. This study revealed that the earliest period of sexual resumption was week two and the latest was 13 weeks. Look this with your definition of early which is of less than 6 weeks.

Reviewer #2: all coments have ben addressed.only concers about format of article. maybe it would be better changed to journal format.

Reviewer #3: Comment1: Best evidences about the best timing for “sexual intercourse start” after delivery are lacking. Some say after two weeks, others say 6 weeks and some others blindly counsel if the mother is ready for the intercourse, she can enjoy (unknown time). So, please, incorporate your best advice (with strong recommendations) in the methodology part!

Comment2: Your title is too sensitive. Asking and getting true information about the issue may be questionable. Had been self-administered, it would be good. So, please, explain when (entry or exit time) and how data collectors interviewed the study participants.

Comment3: Comment on your study population

Your study population is all postpartum women who resumed sexual intercourse and attended the 14th immunization schedule at the setting. Why you didn’t take the 6th immunization schedule for the interview? Because at this time you could get whether or not the study participants resumed their sexual intercourse. The longer the time (14th schedule), the higher the recall bias will be.

Comment4: Comment on your sample size

Your sample size is small. It was good if you were using degree of precision less than 5% because the proportion you used is 73.4%, rule of thumb directs to use either 3% or 4% of degree of precision for such proportion. So that you would increase your sample size and generalizability as well. So, if any reason to use your small sample size you can tell your readers. I know my question is the proposal stage question!

Comment5: On the data processing and analysis section

The author used the P-value ≤ 0.2 in the crudes odds ratio to declare a variable as significant. Please, indicate or cite where you got this cut of point!

Comment6: Comments on the result section, socio-demographic part: The author said ‘About 199(62.4%) participants were from urban residence and 120(37.6% were from rural residence’… The reddened one can be removed because everything is found in the table.

Comment7: 5.2 Obstetrics and Reproductive related characteristics: The author used a sentence ‘About 53 (28.2%) of participants reported a history of abortion and of this nearly half, 46.8 (86.8%) reported once’… The sentence is not clear, especially at the reddened one!

Comment8: Comment on your sentence structure

Please, revise your grammar, subject-verb agreement, active and passive voice of sentences (example: Were resumed, were practiced, number of study participants were reported etc…) had made your sentences vague. Please, try to revise your manuscript with language expertise!

Comment9: In the discussion section, paragraph 5; the author said ‘But earlier than studies reported from Malaysia (7 weeks), Nigeria (8 weeks), and Poland (10 weeks) [9, 17, 18] and these discrepancies might be since postpartum sexuality counseling received from health care providers better than in this study.’ This sentence seems to be paradoxical. The earliest practice is 2 weeks in you study and 7 and above in the comparison studies but your logical explanation is not clear. If am not mistaken, please, correct!

Comment10: On the decimal points

Please, try to use only two digits after the decimal point to make the numbers attractive to be read... Scan your full length manuscript for the problem!

Commen11: Confidence intervals

I have seen some of your confidence intervals are wide which could be due to your small sample size… What is the recommended difference (scientific logic) between the upper and lower boundaries of the CI?

Comment12: Limitation

Don’t you think that the recall bias is there? Because you got the study participants at the 14th week postpartum, they might have forgotten whether or not they practiced sexual intercourse at 2, 3 4…. 13 weeks

7. PLOS authors have the option to publish the peer review history of their article (what does this mean?). If published, this will include your full peer review and any attached files.

Reviewer #1: No

Reviewer #2: **Yes: **Leila Amiri Farahani

Reviewer #3: No

---

## [Author Response · Author response to Decision Letter 1]

27 Jan 2021

Article Title: Early resumption of sexual intercourse and associated factors among married postpartum women at Jimma zone Public Hospitals, South West Ethiopia: A cross-sectional study

1. Response to Reviewer One 

First, we want to appreciate your smart comments and suggestions, and said these here are our concerns and with some clarification on comments given:

Generally comments

1. Generally, comments on grammar, spelling, and subject-verb agreement= revised 

 2. Yes, this study was done on mothers visited for vaccination of their child at 14 weeks.

• In the beginning, I intended to see a number of mothers resumed sexual intercourse during recommended and unrecommended period. We used 14 weeks because it is 3rd EPI schedule in Ethiopia and to access postpartum women easily.

• This study included mothers who visited immunization centers and women visiting other MCH services who were not approached by this study and we added them as a limitation of this study.

Methods sections.

• Many improvements have been done and the population also merged into one.

Result sections

• Amended

 Thank you very much.

 With best regards!

2. For the second reviewer & many thanks for your suggestions, comments, and concerns:

• We have modified the language clarity issue accordingly.

• From the previous comments, we have made many changes to the document to have the PLOS-ONE journal format. 

 Thank you in advance.

 With best regards! 

3.For the second reviewer.Many thanks for your suggestions, comments, and concerns. 

As per the comments given, here are a response to your comments and concerns 

1. Comment 1: my evidence for the best time for sexual resumption is a published research article cited by [1,7,8,9,16,17 ].The best time to resume sexual resumption is after six weeks postpartum. It is a recommendation of many scholars, and also we included operational definitions in the methods part.

2. Yes, it is a sensitive topic but silent adverse health impact on the mother and child health also on the family at all. As you said, had it been a self-administered question, it was nice but this study is facility-based and almost 26% of participants are illiterate.

3. The reason why I took the 14 weeks rather than 6 weeks (2st EPI schedule in Ethiopia) is my intention at first was to see a percentage of women resumed during recommended vs unrecommended period. Balancing those resumed before and after six weeks. Also is selected because 14 weeks is the 3rd EPI schedule in Ethiopia to access all the postpartum women.

4. Sample size issue we used 73% proportion which was taken from the study done in Addis Ababa. This study is the most recent study done in Ethiopia.

5. P≤2 this cut of a point is taken from reference number [5]

6. Modified according to the comment give, and the place of residence is not mentioned under the table so we wrote it in the form text.

7. Modified accordingly 

8. We revised the whole manuscript with language expertise.

9. The explanations issue was modified

10. Decimal point corrected throughout the document

11. Confidence interval issue, we will read it but still, we did not get the exact cut-off point for it.

12. This study is not free from recall bias. However, to minimize recall bias careful design was considered and data collectors asked them freely to remember it.

Thank you very much.

 With best regards!

---

## [Decision Letter · Decision Letter 2]

15 Feb 2021

Early resumption of postpartum sexual intercourse and its associated risk factors among married postpartum women who visited public hospitals of Jimma zone, Southwest Ethiopia: A cross-sectional study

PONE-D-20-25368R2

Dear Dr. Gadisa,

We’re pleased to inform you that your manuscript has been judged scientifically suitable for publication and will be formally accepted for publication once it meets all outstanding technical requirements.

Kind regards,

Nülüfer Erbil, Ph.D, Prof.

Academic Editor

PLOS ONE

---

## [Editor Report · Acceptance letter]

18 Mar 2021

PONE-D-20-25368R2 

Early resumption of postpartum sexual intercourse and its associated risk factors among married postpartum women who visited public hospitals of Jimma zone, Southwest Ethiopia: A cross-sectional study. 

Dear Dr. Gadisa:

I'm pleased to inform you that your manuscript has been deemed suitable for publication in PLOS ONE. Congratulations! Your manuscript is now with our production department. 

Kind regards, 

on behalf of

Dr. Nülüfer Erbil 

Academic Editor

PLOS ONE